# 5-Aminolevulinic Acid (5-ALA)-Induced Protoporphyrin IX Fluorescence by Glioma Cells—A Fluorescence Microscopy Clinical Study

**DOI:** 10.3390/cancers14122844

**Published:** 2022-06-08

**Authors:** Simone Pacioni, Quintino Giorgio D’Alessandris, Stefano Giannetti, Giuseppe Maria Della Pepa, Martina Offi, Martina Giordano, Valerio Maria Caccavella, Maria Laura Falchetti, Liverana Lauretti, Roberto Pallini

**Affiliations:** 1Department of Neuroscience, Università Cattolica del Sacro Cuore, 00168 Rome, Italy; s.pacioni@tiscali.it (S.P.); stefano.giannetti@unicatt.it (S.G.); martinaoffi.mo@gmail.com (M.O.); marty-gio@live.it (M.G.); valeriomaria.caccavella01@icatt.it (V.M.C.); liverana.lauretti@unicatt.it (L.L.); 2CNR-Institute of Cell Biology and Neurobiology (IBCN), 00015 Rome, Italy; marialaura.falchetti@cnr.it; 3Operative Unit of Neurosurgery, Fondazione Policlinico Universitario A. Gemelli IRCCS, 00168 Rome, Italy; quintinogiorgio.dalessandris@policlinicogemelli.it (Q.G.D.); giuseppemaria.dellapepa@policlinicogemelli.it (G.M.D.P.)

**Keywords:** high-grade glioma, low-grade glioma, fluorescence-guided surgery, 5-ALA, blood–brain barrier

## Abstract

**Simple Summary:**

5-aminolevulinic acid (5-ALA)-induced PpIX fluorescence is used in neurosurgery for intraoperative identification of high-grade glioma tissue. In this paper, using a fluorescence microscopy analysis on human tumor specimens, we assessed the actual number of fluorescence-positive tumor cells both in low-grade and high-grade glioma, and the ability of 5-ALA to cross the blood–brain barrier (BBB). We found that in high-grade gliomas, 32.7–75.5 percent of cells display 5-ALA induced PpIX fluorescence, whereas in low-grade gliomas the tumor cells did not fluoresce following 5-ALA. Immunofluorescence for BBB components suggested that 5-ALA does not cross the un-breached BBB. These findings are of crucial importance in planning neurosurgical resection of gliomas.

**Abstract:**

5-aminolevulinic acid (5-ALA)-induced PpIX fluorescence is used by neurosurgeons to identify the tumor cells of high-grade gliomas during operation. However, the issue of whether 5-ALA-induced PpIX fluorescence consistently stains all the tumor cells is still debated. Here, we assessed the cytoplasmatic signal of 5-ALA by fluorescence microscopy in a series of human gliomas. As tumor markers, we used antibodies against collapsin response-mediated protein 5 (CRMP5), alpha thalassemia/mental retardation syndrome X-linked (ATRX), and anti-isocitrate dehydrogenase 1 (IDH1). In grade III–IV gliomas, the signal induced by 5-ALA was detected in 32.7–75.5 percent of CRMP5-expressing tumor cells. In low-grade gliomas (WHO grade II), the CRMP5-expressing tumor cells did not fluoresce following 5-ALA. Immunofluorescence with antibodies that stain various components of the blood–brain barrier (BBB) suggested that 5-ALA does not cross the un-breached BBB, in spite of its small dimension. To conclude, 5-ALA-induced PpIX fluorescence has an established role in high-grade glioma surgery, but it has limited usefulness in surgery for low-grade glioma, especially when the BBB is preserved.

## 1. Introduction

In neurosurgical procedures, 5-aminolevulinic acid (5-ALA)-induced PpIX fluorescence has been used to intra-operatively identify the tumor cells of high-grade gliomas [1,2,3]; 5-ALA is a biochemical precursor from the heme group [4]. Exogenous administration of 5-ALA results in intracellular concentration of endogenous porphyrins, mainly protoporphyrin IX (PpIX) [5,6]. Depending on the rate of catabolism of PpIX, the tissue accumulates this porphyrin. In several neoplastic tissues, an altered metabolism of heme results in increased accumulation of porphyrins [7]. In patients undergoing surgery for malignant glioma, 5-ALA (Gliolan^TM^, Medac GmbH, Wedel, Germany) is administered before general anesthesia, leading to an accumulation of fluorescent PpIX in the tumor cells. Increased PpIX accumulation in the tumor in comparison with a normal brain makes the tumor identifiable during surgery [8]. However, the issue of whether 5-ALA-induced PpIX fluorescence consistently stains all the tumor cells is still debated. Given that glioblastoma (GBM) is a highly heterogeneous malignancy [9,10,11], it is conceivable that some populations of tumor cells are able to avoid 5-ALA labeling due either to a reduced up-take of 5-ALA or to an accelerated catabolism of PpIX or to both [12,13]. Several studies suggest that a consistent percentage of GBM cells in the tumor are not labelled by 5-ALA [14,15]. The fluorescent and non-fluorescent tissue, as they appeared under the operating microscope, contained tumor cells on histology in 95–95.7% and 74–87.5% of samples, respectively [15,16]. However, only a few studies addressed this issue by fluorescence microscopy and, above all, none of these studies used patient-derived specimens. One single work used 5-ALA-induced PpIX fluorescence to retrieve the tumor cells from surgical specimens by cell sorting, suggesting that 5-ALA induced PpIX fluorescence can be used to selectively isolate and analyze cells from human tumor samples [14]. Other than the ability of the tumor cells to avoid 5-ALA labeling, the question of whether 5-ALA does cross the un-breached blood–brain barrier (BBB) still remains unanswered [4]. This issue is clinically relevant because demonstrating any fluorescence in low-grade gliomas, where the tumor cells are able to invade the brain along perivascular spaces without disrupting the BBB, might extend the use of 5-ALA in these tumors. The centerpiece of our paper, that is, the main distinction between the present article and the apparatus/method used by other groups, is visualizing the fluorescence of PpIX by confocal microscopy on patient-derived specimens.

## 2. Materials and Methods

### 2.1. Drug Preparation

Patients were assigned to 5-ALA (Gliolan^®^, Medac GmbH, Wedel, Germany) for fluorescence-guided tumor resection. These patients were scheduled to receive freshly prepared solutions of 5-ALA (20 mg/kg bodyweight) orally 3 h (range 2–5) before induction of anesthesia [8]. Solutions were prepared by dissolving the contents of a bottle (1.5 g) in 50 mL of drinking water. All patients were pre-treated with 12 mg/day of dexamethasone for at least 2 days before administration of 5-ALA.

### 2.2. Clinical Material

Tumor specimens were obtained during craniotomy surgery. This study was approved by the Institutional Ethics Committee of Fondazione Policlinico Gemelli, Università Cattolica, Rome (Prot. 41474/17) and was conducted in accordance with the principles set forth in the World Medical Association Declaration of Helsinki and later amendments. All patients provided written informed consent to the study. Surgical microscopes with a dedicated fluorescence filter (Carl Zeiss, Oberkochen, Germany; Leica microsystems, Wetzlar, Germany) were used for fluorescence-guided tumor resection. Tumor regions showing either hyper-intense FLAIR signal or various degrees of enhancement on gadolinium magnetic resonance (Gd-MR) were located during surgery using a neuro-navigational system (SthealthStation, Medtronic, Minneapolis, MN, USA) in low-grade and high-grade gliomas, respectively. For each patient, analyses were performed on the sample showing the least degree of hemorrhage and necrosis. Twenty-nine patients were enrolled in this study (Table 1).

### 2.3. Microscopic Detection of 5-ALA

Induced PpIX fluorescence can be excited in different spectral regions (Appendix A). PpIX mainly absorbs blue-violet light around 405 nm, corresponding to the largest absorption peak of the fluorophore, and emits a far-red fluorescence with a main peak localized at around 634 nm and extending to about 720 nm. However, there are other smaller absorption peaks of PpIX, which allow the detection of fluorescence emissions in the range of 625–755 nm [17]. Detection of 5-ALA-induced PpIX fluorescence from glioma specimens was obtained by a laser confocal microscope Olympus FLUOVIEW FV1200 equipped with a high sensitivity GaAsP detector module, which detected PpIX fluorescence tissue emission in the range of 655–755 nm wavelengths due to red-light excitation at 635 nm, thus matching a secondary small absorption peak of PpIX. In the same tissues, a concomitant laser confocal microscope configuration adopted for detection of 4′6-diamidino-2-phenylindole (DAPI; excitation/emission: 405/461 nm) and Alexa Fluor 488 secondary antibody (excitation/emission: 488/520 nm) did not overlap with wavelengths of excitation and emission adopted for 5-ALA detection. We used the 635 nm excitation wavelength for PpIX imaging because, though the PpIX absorption coefficient at this wavelength is smaller than at 405 nm, the emission signal is more stable at 635 nm wavelength, thus resulting in better quality detection (Appendix A). Furthermore, we acquired images maintaining low and constant gain parameters for each confocal acquisition. Since 5-ALA-induced PpIX in vivo staining is very easily degradable, special attention was paid to specimen handling. Within 10 min after acquisition in the operating room, the tumor samples were placed in shielded containers and fixed by immersion in 4.5% formalin for 24 h at 4 °C, post-fixed in 30% sucrose, and sectioned (40–50 μm) by a cryostat. All these procedures were conducted in darkness. For quantitative assessment of cell counts, the FV10-ASW 4.2 Viewer program was used. In each sample, at least 1000 tumor cells were counted in 10 no-superimposing high-power fields.

### 2.4. Lectin Staining and Immunofluorescence

Slices were incubated overnight at 4 °C in PB with 0.3% Triton X-100 and 0.1% NDS with Lectin from Lycopersicon esculentum (tomato) biotin conjugate (1:500; Sigma-Aldrich, St. Louis, MO, USA) together with primary antibodies. Sections were incubated overnight at 4 °C in PB with 0.3% Triton X-100 and 0.1% NDS with Lectin from Lycopersicon esculentum (tomato) biotin conjugate (1:500; Sigma-Aldrich, St. Louis, MO, USA) in combination with other antibodies. The monoclonal antibodies used were as follows: mouse anti-isocitrate dehydrogenase (IDH1; R132H; clone HMab-1, 1:50, Sigma Aldrich, St. Louis, MO, USA); rat anti-Collapsin Response-Mediated Protein 5 (CRMP5, 1:50, Millipore, Billerica, MA, USA); mouse anti-Claudin-5 (1:100; Thermo Fisher Scientific, Waltham, MA, USA); mouse anti-Glucose Transporter GLUT1 antibody (1:100; Abcam, Cambridge, UK); mouse anti-human alpha thalassemia/mental retardation syndrome X-linked (ATRX) antibody (1:100; Atlas Antibodies, Bromma, Sweden); mouse anti-human α Smooth Muscle Actin (Monoclonal Mouse, clone 1A4. 1:100 dilution, Dako); and mouse anti-IDH1 (R132H; clone HMab-1, 1:50, Sigma Aldrich, St. Louis, MO, USA). The oolyclonal antibodies used were as follows: rabbit anti-Glucose Transporter GLUT1 antibody (1:200; NovusBio, Centennial, CO, USA); rabbit anti-ZO-1 (1:100; Thermo Fisher Scientific, Waltham, MA, USA); and goat anti-GFAP (1:1000; Thermo Fisher Scientific, Waltham, MA, USA). Slices were rinsed and incubated in PB containing 0.3% Triton X-100 with Alexa Fluor 488 donkey anti-mouse antibody (1:500; Life Technologies, Monza, Italy) for 2 h at RT. Before mounting, slices were incubated for 10 min with DAPI. Immunofluorescence was observed and acquired with a laser confocal microscope Olympus FLUOVIEW FV1200. The following objective lenses were used: Olympus UPlanSAPO 40×/0.95 Oil, Olympus PlanApo N 60×/1.42 Oil, and UPlan FL N 40× A/1.30 Oil. All images were acquired in Z-stack (0.5 μm step size) by scanning several (8–10) Z-planes for each field of acquisition. Images (generally 1600 × 1600 pixels format) were acquired at pinhole size 1 AU, 1–gain, 5 Line-Average, 80.0 us/Pixel scanning speed, and with specific laser power, depending on the type of laser: laser 405 nm: 15% power; laser 488: 25% power; laser 635: 30% power. All confocal images obtained from different samples were acquired using identical confocal settings.

### 2.5. Antigen Retrieval and Auto-Fluorescence Removal

To retrieve CRMP5 antigen and to reduce the masking effect of formalin fixation, the tumor tissues were treated as follows. Sections were immersed for 45′ in a Citrate Buffer antigen retrieval Solution (CBS, composed by 10 mM citric acid, 0.05% Tween 20, pH 6.0), preheated in a water bath at 95–98 °C, and then removed from the bath allowing slides to cool for 30 min at RT. Sections were washed three times in phosPB (10′ each) and incubated with primary antibodies, as described. To remove human-tissue auto-fluorescence caused by the presence of lipofuscin granules in the central nervous system, at the end of the immunofluorescence procedures, sections were treated with Sudan Black B. Following incubation in DAPI (Sigma-Aldrich, Cat. #D9542, concentration 1:4000) for 10 min, sections were rinsed three times in a phosphate buffer (PB), and then immersed in 0.3% Sudan Black B (Sigma Aldrich) in ethanol 70% for 25–30 s at room temperature, followed by another 3 consecutive washing steps in PB (10′ each).

## 3. Results

### 3.1. Fluorescence Microscopy for 5-ALA and Lectin Co-Staining

Direct examination of frozen tumor sections avoiding any manipulation allows the best detection of 5-ALA fluorescent signals (Figure 1). Procedures such as co-staining with lectin and, above all, immunofluorescence for tumor markers, consistently reduced the degree of 5-ALA-induced PpIX fluorescence. In grade III and IV gliomas, the fluorescent signal localizes in the cell cytoplasm of cells as a dotted staining. The fluorescent signal is detected in 58.6 ± 10.7 percent (mean ± sd) of the total cell population, including the non-neoplastic cells. Each labeled cell contained from 7 to 142 fluorescent dots (38.5 ± 38.1, mean ± sd) (Figure 1). Using the direct technique of observation, however, we cannot distinguish the neoplastic tumor cells from non-neoplastic cells, such as tumor-associated macrophages or infiltrating lymphocytes, since atypical nuclei represent an unreliable marker of malignancy. Staining of tumor sections with lectin, in spite of losing some fluorescence intensity, revealed that at 4–6 h after 5-ALA administration, fluorescent cells are still present in the vessel lumen of gliomas of any grade (Figure 2). The endothelial cells, which were in contact with circulating 5-ALA, had weak or no fluorescent signals, suggesting that these cells either do not up-take the dyer or are able to metabolize it very rapidly. Activated pericytes, which were stained with the anti-αSMA antibody, did not fluoresce following 5-ALA, suggesting that the fluorescence signal along perivascular spaces was due to infiltrating tumor cells (Appendix A). Outside the vessels, fluorescent cells were not detected in grade II gliomas, whereas fluorescent cells were found in all malignant gliomas. In grade II gliomas, only a few weakly fluorescent cells inside the vessels were seen (Figure 2). In grade III–IV gliomas, the tumor cells that were in close spatial relationship with the vessel wall fluoresced brightly upon 5-ALA (Figure 2). The vessel wall was often discontinuous on lectin staining.

### 3.2. Immunofluorescence Microscopy

In order to determine the actual percent of tumor cells labeled by 5-ALA, we used the anti-CRMP5 antibody for selective staining of the tumor cells. Differently from in vivo models, where the tumor cells can be labeled and traced through the brain, in human specimens the tumor cells are difficult to stain with specific markers [18]. The antibody against CRMP5 was recently proposed as a selective tumor marker for glioma cells [19,20]. Previously, our group validated this antibody on patient-derived glioma stem-like cells and on surgical specimens of glioma with mutant IDH1/2, in which we showed a co-staining of the tumor cells with anti-IDH1/2 and anti-CRMP5 antibodies [21,22]. Here, double staining of IDH wt GBM with anti-CRMP5 and with anti-ATRX antibody, often used in combination with other glioma markers, revealed loss or hypo-expression of this marker (Appendix A). As mentioned above, manipulations of the tissue specimen for immunofluorescence reduce 5-ALA-induced PpIX fluorescence to the point that the two techniques cannot be used simultaneously on the same section. However, when applied separately on adjacent tumor sections, these techniques allow the approximation of the percent of CRMP5-positive tumor cells that do fluoresce upon 5-ALA administration. In low-grade gliomas, the tumor cells that were stained with CRMP5 did not fluoresce upon 5-ALA. Differently, 5-ALA-induced PpIX fluorescence was seen in 32.7 ± 13.3 and 75.5 ± 9.8 percent (mean ± sd) of CRMP5-positive tumor cells in grade III gliomas and GBM, respectively (Figure 3).

We then investigated the relationship between 5-ALA-induced PpIX fluorescence and BBB integrity using immunofluorescence with antibodies directed against Glut1 and ZO-1 [22]. As expected, in low-grade gliomas, both the Glut1 and ZO-1 staining were well represented, suggesting an intact BBB (Figure 4). Those regions of high-grade gliomas that did not fluoresce intra-operatively showed an abnormal but still present BBB, as suggested by preserved Glut1 and ZO-1 staining. In those areas of high-grade gliomas with Gd enhancement on MRI that displayed bright fluorescence after 5-ALA, the BBB was greatly disrupted (Figure 4). We then combined 5-ALA-induced PpIX fluorescence with Glut1 immunofluorescence—the one meant for studying the BBB that alters 5-ALA-induced PpIX fluorescence to a minor degree (Figure 5). Again, in low-grade gliomas, 5-ALA-induced PpIX fluorescence was detected in only a few intravascular cells while the vascular Glut1 staining was strong. Conversely, in high grade gliomas, 5-ALA-induced PpIX fluorescence was bright in perivascular cells and still detected within the tumor where the Glut1 staining was discontinuous (Figure 5). These results suggest that 5-ALA-induced PpIX fluorescence is closely related with BBB disruption.

## 4. Discussion

The present study is the first one showing the 5-ALA signal of human gliomas under confocal microscopy. The results can be summarized as follows: (1) in grade III–IV gliomas, the signal induced by 5-ALA can be detected in 32.7–75.5 percent of CRMP5-expressing tumor cells; (2) in the low-grade gliomas (grade II), the CRMP5-expressing tumor cells do not fluoresce following 5-ALA; and (3) 5-ALA-induced fluorescence was not detected in the areas with un-breached BBB. 

Direct fluorescence microscopy of frozen sections avoiding immunostaining allows optimal detection of the fluorescent signal induced by 5-ALA. This signal, which is located in the cell cytoplasm and appears as a dotted staining, can be due to an accumulation of PpIX in cell mitochondria [6]. Interestingly, in the HER2 transformed breast epithelial cells, the accumulated PpIX is mainly confined to the mitochondria [23]. In the present study, the number of fluorescent dots varied considerably among individual cells and this variation may account for under-detection of the tumor cells during surgery. In a clinical trial studying 5-ALA-induced PpIX fluorescence in high-grade gliomas, 5-ALA fluorescence predicted the presence of tumors [24]. However, over 60% of glioma samples with positive histology did not fluoresce. Vague fluorescence or lack of fluorescence in brain tumors may be due to infiltrative tumors invading healthy tissue and/or the inability of 5-ALA to cross an intact BBB [25,26]. Technical issues, such as the reduction of fluorescence due to photobleaching, cannot be completely overruled; however, the true impact of this phenomenon is still debated, as the rate of fluorescence decay under microscope light is low [27]. On the other hand, it has been demonstrated previously that PpIX photobleaching may significantly impact interpretation of the findings [28]. In addition, it was also our personal observation that PpIX photobleaches quickly under the confocal excitation laser, especially at 405 nm, so that the sample could only be scanned a few times before the brightness diminished substantially. To address these issues, we used a rigid protocol for handling specimens and adopted the 635 nm excitation wavelength, which allowed us to obtain a more stable signal (see Methods and Appendix A). Real-time intraoperative confocal imaging could be of help in reducing tissue manipulation and improving fluorescence quality [29].

One issue that is relevant on clinical grounds is whether 5-ALA is up-taken by the tumor cells in low-grade glioma. Jaber et al. reported that 21.6% of histologically confirmed low-grade gliomas fluoresced upon 5-ALA [30]. However, they also described a weak contrast-enhancement on the MRI of these tumors, suggesting that fluorescence in low-grade gliomas may result from early angiogenic changes. ALA is a small, water-soluble molecule; however, its ability to pass the BBB is still subject to dispute [4,25]. It is generally believed that ALA for itself does not cross the intact BBB. Possibilities that could account for the low/no fluorescence in the grade I–II glioma could be the low metabolic activity and/or the potential for 5-ALA to not cross the BBB. Two in vivo studies reported completely different results [31,32]. Olivo et al. administered 5-ALA (100 mg/kg iv) to normal and tumor-bearing rabbits and studied the distribution of fluorescence in the brain by fluorescence microscopy [31]. They showed that ALA is able to cross an intact BBB even in regions of the brain that are remote from the meninges or choroid plexus. Conversely, Hebeda et al., who used fluorescence imaging for the localization of porphyrins in the rat brain tumor model after ALA administration, did not detect any fluorescence in the striatum contralateral to the tumor [32]. Within the tumor, the fluorescence signal showed large variations in intensity. In perivascular regions, the cytoplasm of the tumor cells had pronounced fluorescence with demarcation of the islands of tumor growth.

During surgery, the brightness of fluorescence may differ from zone to zone of the same tumor, depending, among other factors, on the cellularity. However, our results showed that even in tumor zones of high cellularity and no necrosis, there may be a subset of tumor cells which does not fluoresce under confocal microscopy after 5-ALA administration. From a neurosurgical standpoint, our paper does not change the daily work in the operating room. Neurosurgeons know that not all the tumor cells of high-grade glioma fluoresce upon 5-ALA administration and that the tumor cells of low-grade glioma do not fluorescence at all. In spite of this, the direct visualization of 5-ALA positive tumor cells combined with immunofluorescence, given the highly sensitive conditions of confocal microscopy, may provide useful data on the distribution of this dye relative to cell type, vascularity, and BBB integrity.

Interestingly, Olivo et al. found the fluorescent signal around the brain vessels at 3 h after 5-ALA administration, whereas the endothelial cells were not mentioned [31]. In the work by Hebeda et al., the thickness of the fluorescent vessel wall suggests either that the pericytes do fluoresce [32] or that the tumor cells invaded perivascular spaces. However, apart from the vessel wall, whose fluorescence signal may be affected by the amount of 5-ALA administered and by the rate of its metabolism into endothelial cells, our results on human samples of glioma are consistent with those of the in vivo study by Hebeda et al. [32]. That is, in human gliomas, 5-ALA-induced PpIX fluorescence seems to be linked with disruption of the BBB. A gradient of fluorescent signals was noted, whereby the tumor cells that invade perivascular spaces fluoresce brightly, whereas those located at a distance from the vessels do not show 5-ALA fluorescence.

## 5. Conclusions

The fluorescent 5-ALA signal can be detected in up to 75% of high-grade glioma cells. Conversely, in low-grade gliomas, especially when the BBB is intact, the tumor cells do not show 5-ALA-induced PpIX fluorescence. Our findings may be useful to neurosurgeons during 5-ALA guided removal of gliomas.

## Figures and Tables

**Figure 1 cancers-14-02844-f001:**
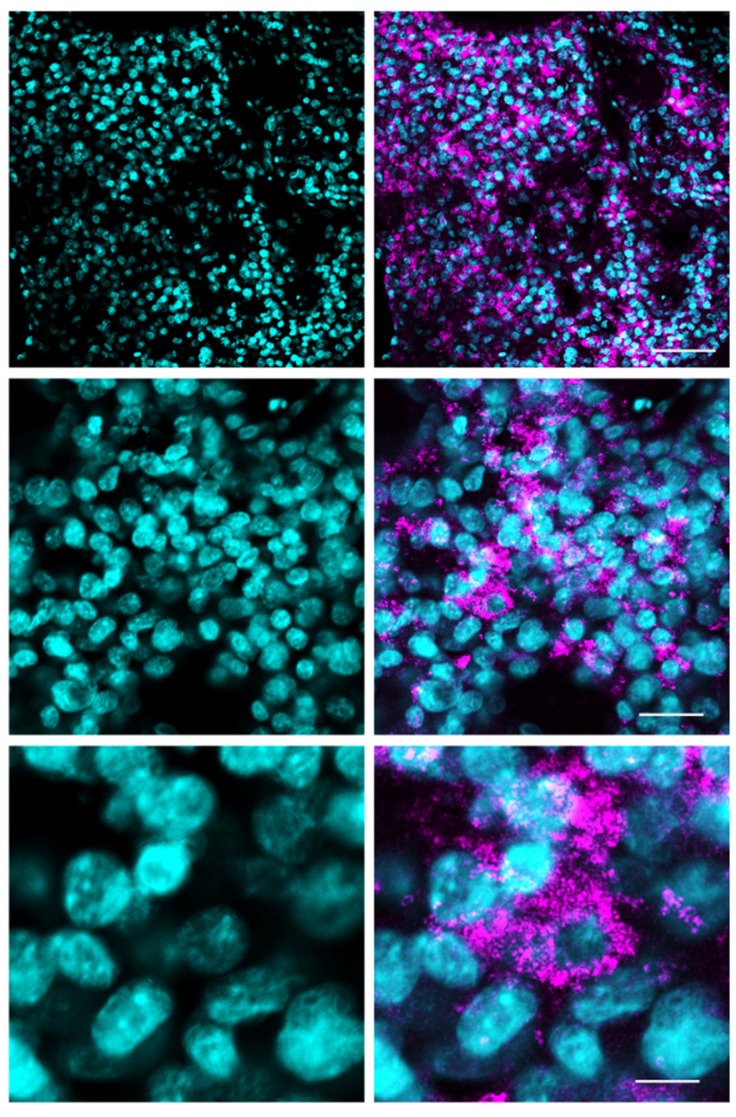
Fluorescence microscopy of a 5-ALA fluorescent GBM tumor (Case #16 on Table 1). Frozen sections counterstained with DAPI. Scale bars, 70 μm (upper panels), 25 μm (middle panels), and 10 μm (lower panels). Objective lens, Olympus UPlanSAPO 40×/0.95 Oil.

**Figure 2 cancers-14-02844-f002:**
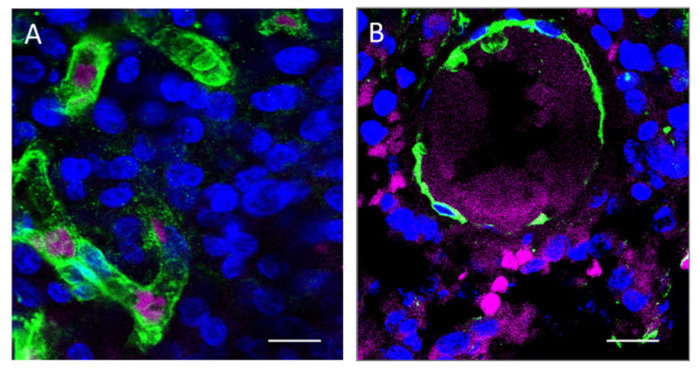
Fluorescence microscopy for 5-ALA and lectin. (**A**) In low-grade gliomas (Case #2 on Table 1), 5-ALA fluorescent cells (purple) can be found occasionally within vascular structures (green). Scale bar, 20 μm. (**B**) In high-grade gliomas (Case #5 on Table 1), several 5-ALA fluorescent cells (purple) are seen outside blood vessels (green) whose wall appears discontinuous. Scale bar, 30 μm. Objective lens, Olympus PlanApo N 60×/1.42 Oil.

**Figure 3 cancers-14-02844-f003:**
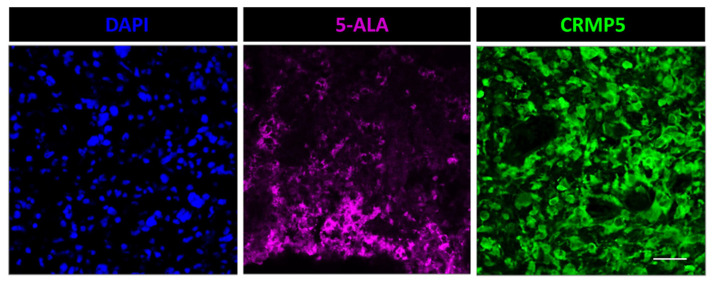
Fluorescence microscopy for 5-ALA and immunofluorescence with anti-CRMP5 of adjacent sections in a case of GBM (Case #12 on Table 1). About 60% of the CRMP5 expressing tumor cells (green) do fluorescence after 5-ALA (purple). Scale bar, 40 μm. Objective lens, Olympus UPlanSAPO 40×/0.95 Oil.

**Figure 4 cancers-14-02844-f004:**
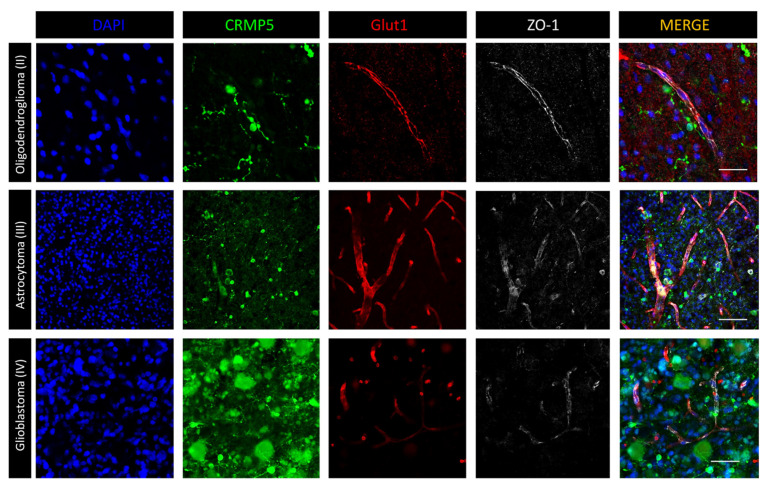
Immunofluorescence with anti-CRMP5 for staining of the tumor cells and with anti-Glut1 and anti-ZO1 for staining of BBB. In low grade gliomas (grade II oligodendroglioma, upper panels; Case #3 on Table 1), a few morphologically differentiated tumor cells do not disrupt the BBB that appears well preserved with maintained Glut1 and ZO1 staining. Scale bar, 50 µm. High grade glioma (grade III astrocytoma, middle panels; Case #5 on Table 1) show an increased density of blood vessels with discontinuous Glut1 and lost of ZO1 dotted staining. Scale bar, 100 µm. In GBM (grade IV astrocytoma, lower panels; Case #26 on Table 1), the CRMP5 expressing tumor cells are morphologically heterogeneous. The blood vessels show highly disrupted BBB with losing of Glut1 and ZO1 staining. Scale bar, 40 µm. Objective lens, Olympus UPlanSAPO 40×/0.95 Oil.

**Figure 5 cancers-14-02844-f005:**
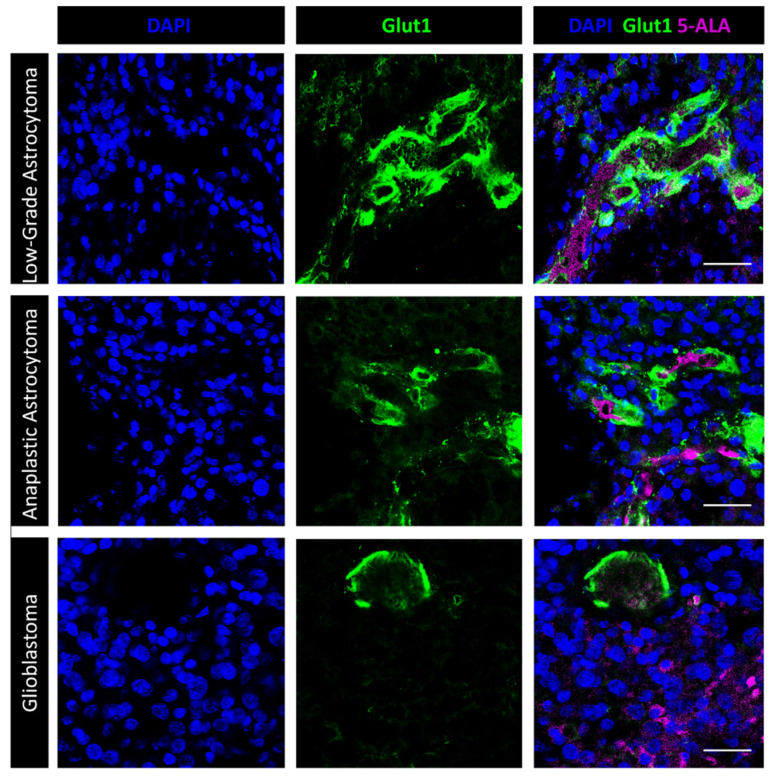
Fluorescence microscopy for 5-ALA and immunofluorescence with anti-Glut1. In low grade glioma (upper panels; Case #2 on Table 1), the dotted 5-ALA fluorescent signal is mostly confined to intravascular cells. A few fluorescent dots can be seen in perivascular cells. Glut1 is highly expressed by the endothelial cells. Scale bar, 40 μm. In anaplastic glioma (middle panel; Case #4 on Table 1), the Glut1 staining lacks in many regions of the tumor vessels, suggesting BBB disruption. Mosaic and/or perivascular tumor cells show intense 5-ALA fluorescence. Scale bar, 40 μm. Glioblastoma tumors (lower panels; Case #14 on Table 1) show several 5-ALA fluorescent cells even at distance from vessels with discontinuous Glut1 staining. Scale bar, 40 μm. Objective lens, Olympus PlanApo N 60×/1.42 Oil.

**Table 1 cancers-14-02844-t001:** General features of tumor specimens.

Case	Age/Sex	Tumor Location	No. Samples	Histology (WHO Grade)	Molecular Profile	Comment
1	40/M	R frontal	3	Astrocytoma (II)	IDH mutant	
2	34/M	R frontal	4	Oligodendroglioma (II)	IDH mutant	
3	59/M	R parietal	4	Oligodendroglioma (II)	IDH mutant	Recurrent
4	49/M	L frontal	2	Astrocytoma (III)	IDH mutant	
5	40/F	R frontal	3	Astrocytoma (III)	IDH wild type	Recurrent
6	44/F	L frontal	4	Oligodendroglioma (III)	IDH mutant	
7	29/M	R frontal	2	Oligodendroglioma (III)	IDH mutant	
8	42/M	L temporal	3	Oligodendroglioma (III)	IDH mutant	
9	48/M	R occipital	2	Glioblastoma (IV)	IDH1 wild type, EGFRvIII +	Recurrent
10	43/M	R temporal	1	Glioblastoma (IV)	IDH wild type	
11	65/M	L frontal	1	Glioblastoma (IV)	IDH mutant	
12	72/F	R temporal	2	Glioblastoma (IV)	IDH1 wild type, EGFRvIII +	
13	54/M	R parietal	1	Glioblastoma (IV)	IDH1 wild type, EGFRvIII +	
14	68/F	R temporal	3	Glioblastoma (IV)	IDH1 wild type, EGFRvIII +	
15	76/F	R frontal	1	Glioblastoma (IV)	IDH wild type	
16	85/F	L temporal	2	Glioblastoma (IV)	IDH wild type	
17	50/M	L temporal	3	Glioblastoma (IV)	IDH wild type	Recurrent
18	75/M	R parietal	1	Glioblastoma (IV)	IDH wild type	
19	48/M	L frontal	3	Glioblastoma (IV)	IDH mutant	Secondary
20	58/F	L frontal	2	Glioblastoma (IV)	IDH wild type	Recurrent
21	43/F	R frontal	1	Glioblastoma (IV)	IDH wild type	
22	34/F	R frontal	3	Glioblastoma (IV)	IDH mutant	Secondary
23	78/M	L temporal	2	Glioblastoma (IV)	IDH wild type	
24	54/M	R temporal	1	Glioblastoma (IV)	IDH wild type	
25	55/M	R frontal	1	Glioblastoma (IV)	IDH wild type	
26	70/M	R temporal	3	Glioblastoma (IV)	IDH wild type	
27	68/M	R frontal	1	Glioblastoma (IV)	IDH wild type	

IDH, isocitrate dehydrogenase; EGFRvIII, epidermal growth factor receptor variant III.

## Data Availability

Source data are available from the Corresponding Author upon request.

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
