# Peer review of "5-Aminolevulinic Acid (5-ALA)-Induced Protoporphyrin IX Fluorescence by Glioma Cells—A Fluorescence Microscopy Clinical Study"

_cancers, 2022, doi:10.3390/cancers14122844_

Round 1

Reviewer 1 Report

I have seen now the revised version of the manuscript. Some of my methodological concerns remain, especially the concern with respect to the tissue harvetsing site. The authors wrote that they took the sample either from the GD-enhancing area (I guess in HGGs) or from the FLAIR-hyperintense area (I guess in LGGs), guided by neuronavigation. In the table, it can be found that more than one tissue sample was talen in the majority of patients (up to 4). Which sample then was taken for microscopic investigation? If all samples were investigated: Did the authors found differences between the sample in one tumor and how the calculated the percentages of stained cells in those cases? Which attempts were made to control a possible bias?

Author Response

- I have seen now the revised version of the manuscript. Some of my methodological concerns remain, especially the concern with respect to the tissue harvetsing site. The authors wrote that they took the sample either from the GD-enhancing area (I guess in HGGs) or from the FLAIR-hyperintense area (I guess in LGGs), guided by neuronavigation.

Response: This Reviewer is right. In HGGs, we took samples from the Gd-enhancing area, while in the LGG, we took samples from the FLAIR-hyperintense area, guided by neuronavigation. This notation was added in Material and Method section of revised manuscript (page 3, lines 91-92).

- In the table, it can be found that more than one tissue sample was talen in the majority of patients (up to 4). Which sample then was taken for microscopic investigation? If all samples were investigated: Did the authors found differences between the sample in one tumor and how the calculated the percentages of stained cells in those cases? Which attempts were made to control a possible bias?

Response: Not all samples were investigated by confocal microscopy. In each case, we chose the best preserved sample,i.e.. that one showing  little or no haemorrhages and necrosis. This detail was added in Material and Method section of revised manuscript (page 3, lines 92-93).

Reviewer 2 Report

The authors have addressed some points however not fully. Reading reviewer comments and responses to R2 and R4 was especially interesting and clarified some points regarding the manuscript. There are further points that need to be addressed.

1.     R2 Q3 and Q14 needs further clarification. Authors should report the number of observations for each of their statistical calculations. 

“The fluorescent signal is detected in 58.6 + 10.7 percent (mean + sd) of total cell population, including the no-203 neoplastic cells” – 1) from how may total FOV, 2) from how many specimens, 3) from how many patients these calculations were derived from? What are the absolute numbers of cells counted?

2.     R2 Q5 needs further clarification. In line with R4 Q4, the authors need to expand on methods. In particular, did you scan only 1 Z plane or several? What was the gain and laser power settings, scanning speed etc? Was the laser intensity and gain set similarly in all specimens and if Garde 2 tumors with lowest grade fluorescence were chosen as the lowest gain and laser intensity settings across all experiments. If no, it is also fine, it is just need to be reported. I would assume it is hard to do so as LGG are rare, and to wait for one to set the laser power and gain levels might be practically difficult. Especially considering the fact that the fluorescence signal might have bleached due to handling. 

3.     Article title should be changed. First, add the word “induced” to 5-ala “induced” PpIX fluorescence. Second, the article tile should reflect the fact that patient tissues were used.

4.     R2 Q15. The LGG tumor cannot be regarded as negative control to GBM due to different cellular composition. Ideally, authors should present a figure of a GBM specimen treated and imaged identically to a patient that did not receive 5-ALA. 

5.     R4 Q4 reply to the question should be incorporated to the discussion as it provides important information as to why 635 excitation was used. 

6.     Lines 253-255. “However, when applied separately on adjacent tumor sections, these techniques allow determining the percent of CRMP5-positive tumor cells that do fluoresce upon 5-ALA administration.” – should be edited to “…allow to approximate the percent…”. Since the measurements are of the cells from the adjacent sections. 

7.     Does authors have a figure in support of this claim?

8.     Line 271 “Conversely, in high grade gliomas and GBMs,” – Glioblastoma is a high grade glioma. Please correct this sentence to avoid repetition.

9.     Line 274 – extra letter “f” after ALA.

10.   Line 278, 281. “In low grade gliomas (grade II astrocytoma and oligodendroglioma, upper panels; Case #3 on Table 1),”The legend should be specific to the presented image. Is it astrocytoma or oligodendroglioma? Leave just one.  The same comment also regarding the high grade glioma example. Please leave only one description that is characterizing the image: grade III astrocytoma or oligodendroglioma.

11.   Also, please add to the paper in how many specimens the Glut-1 and ZO-1 analysis was performed?

12.   R2 Q3. “The present study is the first one showing the 5-ALA signal of human gliomas under confocal microscopy.” . The acquisition of confocal microscopy images of the 5-ALA in human gliomas has been reported previously (PMID: 28418534). The authors wrote that they were unable to find data on acquisition with confocal microscopy throughout the paper. The authors should read better, as the whole idea of that paper is visualization of 5-ALA signal with a custom dual-axis confocal microscope. The reviewer is familiar with the authors of that publication and is confident in that they have used dual axis confocal microscope to visualize 5-ALA fluorescence in human glioma specimens. 

13.   Line 294: “5-ALA does not cross the un-breached BBB” – this statement should be more speculative as authors have no data to confirm this with certainty. This could be rephrased as “5-ALA induced fluorescence was not detected in the areas with un-breached BBB.”

Author Response

The authors have addressed some points however not fully. Reading reviewer comments and responses to R2 and R4 was especially interesting and clarified some points regarding the manuscript. There are further points that need to be addressed.

  1. 1.R2 Q3 and Q14 needs further clarification. Authors should report the number of observations for each of their statistical calculations. 

“The fluorescent signal is detected in 58.6 + 10.7 percent (mean + sd) of total cell population, including the no-203 neoplastic cells” – 1) from how may total FOV, 2) from how many specimens, 3) from how many patients these calculations were derived from? What are the absolute numbers of cells counted?

Response: for all calculations, 10 FOVs from one specimen per patient were included in the analysis (see also the revised Methods, page 3, lines 92-93). The statistic here reported was performed on 24 patients. As concerns the other statistics from R2 Q3 (32.7 ± 13.3 and 75.5 ± 9.8 percent) they were performed on 5 and 24 patients, respectively.

  1. R2 Q5 needs further clarification. In line with R4 Q4, the authors need to expand on methods. In particular, did you scan only 1 Z plane or several? What was the gain and laser power settings, scanning speed etc? Was the laser intensity and gain set similarly in all specimens and if Garde 2 tumors with lowest grade fluorescence were chosen as the lowest gain and laser intensity settings across all experiments. If no, it is also fine, it is just need to be reported. I would assume it is hard to do so as LGG are rare, and to wait for one to set the laser power and gain levels might be practically difficult. Especially considering the fact that the fluorescence signal might have bleached due to handling.

Response: we thank the Reviewer for these observations. We have added the technical details for image acquisition in the Methods section, on page 5, lines 182-187, as follows: “All images were acquired in Z-stack (0.5μm step size) by scanning several (8-10) Z-planes for each field of acquisition. Images (generally 1600x1600 pixels format) were acquired at pinhole size 1 AU, 1x gain, 5 Line-Average, 80.0 us/Pixel scanning speed and with specific laser power, depending on the type of laser: laser 405 nm: 15% power; laser 488: 25% power; laser 635: 30% power. All confocal images obtained from different samples were acquired using identical confocal settings”.

  1. 3.Article title should be changed. First, add the word “induced” to 5-ala “induced” PpIX fluorescence. Second, the article tile should reflect the fact that patient tissues were used.

Response: the title was changed as follows: “5-Aminolevulinic acid (5-ALA)–induced protoporphyrin IX fluorescence by glioma cells. A fluorescence microscopy clinical study.”

  1. R2 Q15. The LGG tumor cannot be regarded as negative control to GBM due to different cellular composition. Ideally, authors should present a figure of a GBM specimen treated and imaged identically to a patient that did not receive 5-ALA. 

Response: this Reviewer is right. However, 5-ALA administration at our center is the standard-of-care for GBM surgical removal.

  1. R4 Q4 reply to the question should be incorporated to the discussion as it provides important information as to why 635 excitation was used. 

Response: we have incorporated these responses in the Discussion section, on page 10, lines 333-335.

  1. Lines 253-255. “However, when applied separately on adjacent tumor sections, these techniques allow determining the percent of CRMP5-positive tumor cells that do fluoresce upon 5-ALA administration.” – should be edited to “…allow to approximate the percent…”. Since the measurements are of the cells from the adjacent sections.

Response: we have edited the sentence according to the Reviewer’s suggestion. 

  1. Does authors have a figure in support of this claim?

Response: unfortunately, as disclosed in the paper, immunofluorescence for IDH1 and CMRP5 does not allow to detect 5-ALA-induced fluorescence in the same slice.

  1. Line 271 “Conversely, in high grade gliomas and GBMs,” – Glioblastoma is a high grade glioma. Please correct this sentence to avoid repetition

Response: we have removed “GBMs”.

  1. Line 274 – extra letter “f” after ALA.

Response: removed.

  1. Line 278, 281. “In low grade gliomas (grade II astrocytoma and oligodendroglioma, upper panels; Case #3 on Table 1),”The legend should be specific to the presented image. Is it astrocytoma or oligodendroglioma? Leave just one.  The same comment also regarding the high grade glioma example. Please leave only one description that is characterizing the image: grade III astrocytoma or oligodendroglioma.

Response: we have edited the Figure 4 legend according to the Reviewer’s suggestions.

  1. Also, please add to the paper in how many specimens the Glut-1 and ZO-1 analysis was performed?

Response: the analysis was performed on all specimens.

 R2 Q3. “The present study is the first one showing the 5-ALA signal of human gliomas under confocal microscopy.” . The acquisition of confocal microscopy images of the 5-ALA in human gliomas has been reported previously (PMID: 28418534). The authors wrote that they were unable to find data on acquisition with confocal microscopy throughout the paper. The authors should read better, as the whole idea of that paper is visualization of 5-ALA signal with a custom dual-axis confocal microscope. The reviewer is familiar with the authors of that publication and is confident in that they have used dual axis confocal microscope to visualize 5-ALA fluorescence in human glioma specimens. 

Response: we agree with the Reviewer. Wei et al present an elegant paper in which they set up and validate a dual-axis confocal microscopy tool for real-time imaging of fluorescent gliomas. We have added a reference in the Discussion section: “Real-time intraoperative confocal imaging could be of help in reducing tissue manipulation and improving fluorescence quality” (page 10, lines 335-337). In the present study, however, we performed a thorough confocal microscopy analysis of relationships between fluorescent cells, glioma cells, tumor vasculature and blood-brain-barrier across gliomas of different WHO grades.

 Line 294: “5-ALA does not cross the un-breached BBB” – this statement should be more speculative as authors have no data to confirm this with certainty. This could be rephrased as “5-ALA induced fluorescence was not detected in the areas with un-breached BBB.”

Response: we have rephrased the sentence according to the Reviewer’s suggestion.

This manuscript is a resubmission of an earlier submission. The following is a list of the peer review reports and author responses from that submission.

Round 1

Reviewer 1 Report

On the cellular level, the authors investigated tissue of high- and low-grade gliomas obtained during surgery for staining with 5-ALA. They showed that up to 75% of high grade glioma cells have a fluorescent 5-ALA signal and that no fluorescence can be found in low-grade gliomas.

 I have a methodologic concern:  1- It is well known that during GBM surgery the brightness of fluorescence differ which is related (among other factors) to the cellularity and that we have tumor regions, which do not stain. Therefore, if calculating percentages it appears crucial to ensure that the taken samples are truly comparable.  

 And, from a neurosurgical standpoint, I asked myself if these data could change my daily work. The answer is clearly “no”. It is well known since more than a decade of working with 5-ALA, that many, but not all high-grade gliomas stain, and that true low-grade gliomas normally do not stain (which is the reason why 5-ALA now is used today not only in HGG, but also with good results in lymphomas and metastases, but not in low-grade gliomas). If dealing with a fluorescent high-grade glioma, surgery is proceeded until no fluorescence is left. The fluorescence, which is seen under the operating microscope is somehow the sum of the fluorescence of the cells and it does not matter, if the percentage of staining cells is high or low as long as it can be seen under the operating microscope. I propose the authors more clearly explain the potential benefits of their findings and highlight the truly new aspects.

.

Reviewer 2 Report

This is a very nice manuscript on a pertinent topic. It would be of interest to the cancer researchers, clinical neurosurgeons that use the 5-ala technique. There are several minor and major issues with this manuscript that should be addressed before this paper could be recommended for publication:

  1. The table with the list of tumors seems to be not connected with the manuscript content. How many samples per patient?
  2. The authors should include information on each of the figures regarding which case does each particular image is coming from.
  3. The methodology of a quantitative assessment of cell counts is not reported. What program was used? How many FOV were counted, from how many samples, from how many patients, which tumor types? The data reported below should have methodology clarified:

- “detected in 58.6 + 10.7 140 percent (mean + sd) of total cell population”

- “Each la-141 belled cell contained from 7 to 142 fluorescent dots (38.5 + 38.1, mean + sd)”

- “in 32.7 + 13.3 and 75.5 + 9.8 percent (mean + sd) 190 of CRMP5-positive tumor cells in grade III gliomas and GBM, respectively”

  1. Handling of the specimens from the acquisition in the OR till the staining is not well described, please elaborate, including timing. We have found previously that 5-ala induced PpIX in vivo staining is very easily degradable and requires special specimen handling.
  2. Microscopic detection of 5-ALA is not well detailed, please include objective used, scanning parameters such as pinhole size, Z thickness, etc. This is important to ensure the reproducibility of the results. Especially considering that confocal microscopy is not the most sensitive method for detection of fluorescence due to the pinhole.
  3. The authors have used a 635nm excitation laser for PpIX imaging. Because PpIX absorption coefficient at this wavelength is much smaller than at 405nm, considering how good of a picture they obtained, the authors should have obtained excellent results and a much brighter signal with 405 excitation and detection around 633 nm window. I encourage the authors to include a supplementary image comparing these 2 excitation options on frozen sections.
  4. Does your microscope allow spectroscopic assessment of PpIX signal? It would be important to distinguish characteristic PpIX spectral signature from the lipofuscin signal that frequently contaminates the specimen.
  5. Did you use Sudan Black B on the specimens imaged for PpIX?
  6. Although the authors mentioned that photobleaching cannot be completely eliminated, this PpIX photobleaching phenomenon deserves greater attention. It has been demonstrated previously that PpIX photobleaching may in fact significantly impact interpretation of the findings (PMID: 30135440), especially if the amount of PpIX is low to begin with. In addition, it was also our personal observation, I am sure the authors agree and experienced similarly, that PpIX photobleaches fast under the confocal excitation laser, especially at 405nm, so that the sample could only be scanned several times before the brightness diminishes significantly. Increasing the gain further in such circumstances reveals autofluorescence and noise.
  7. Some of the abbreviations need to be spelled out at the first use (PB, NDS, IDH, EGFR, DAPI.
  8. Specify which DAPI reagent you have used.
  9. 5-ala fluorescence should be changed to PpIX fluorescence or 5-ala induced PpIX fluorescence throughout the manuscript.
  10. “The present study is the first one showing the 5-ALA signal of human gliomas under fluorescence microscopy.” First of all, the authors likely implied “ confocal fluorescence microscopy” This is not correct. 5-ala has been previously studied in human gliomas in several studies (PMID: 28418534).
  11. Reported images and results section should include data on how many specimens these experiments were confirmed.
  12. The authors should include negative controls of tissue imaging without the 5-ALA that demonstrates absence of fluorescence.

Reviewer 3 Report

In this work, the Authors report their experience with a fluorescence microscopy analysis in a heterogeneous population of 27 patients with brain gliomas to assess the presence of 5-ALA fluorescence in tumor cells and its ability to cross an intact blood-brain-barrier. They observed that 32.7-75.5 of grade III-IV tumor cells showed 5-ALA fluorescence, which was absent in all the grade II tumor cells, and finally, that 5-ALA does not cross an intact blood-brain-barrier.

The question the Authors move their investigations from is relevant, as the relationship between 5-ALA fluorescence and the tumor histology in gliomas still has some aspects needing further clarification.

However, the Authors should consider highlighting the implications and impact of the results in their opinion, even potentials: this holds especially true for surgery, as 5-ALA is an eminently operative adjunct. Moreover, statistical analysis should be performed to strengthen their observations, for example, but not limited to, a comparison of proportion between the subpopulation they studied, a comparison between intra-operative and microscopy 5-ALA response, etc. Reviewing from an English native speaker could improve the manuscript readability.

Reviewer 4 Report

5-Aminolevulinic acid (5-ALA)–protoporphyrin IX fluorescence by glioma cells. A fluorescence microscopy study.

Comments to the Editor/Authors:

  1. The manuscript fits well within the scope of Cancers.
  2. The main argument of the manuscript is the authors assessed 5-ALA (i.e., Protoporphyrin IX) in patient-derived series of low to high grade glioma samples via utilizing confocal fluorescence microscopy and whether the negative results (i.e., no fluorescence observed) in the low grade gliomas was reflective of the ability of 5-ALA to cross the blood-brain barrier.
  3. The manuscript is organized in a decent manner, informative, and nicely discussed. However, the manuscript would benefit from some modifications such that additional clarity and direction is provided:
    1. The introduction ought to be put in the context of what other groups/articles have presented in terms of the apparatus/method that they have used to image Protoporphyrin IX in glioma specimens (none in vivo), as one distinction between the authors’ article and the others is here the use of visualizing the fluorescence of Protoporphyrin IX using confocal microscopy on patient-derived specimens. This should be the centerpiece of the paper. By doing so, the overall impact would be significantly greater and placed in the proper context.
    2. Is the Supplementary Figure S1 in line with article discussion and the paper from which the authors obtained the image? In the cited paper, excitation was at 625 nm. However, the authors state that they used an excitation wavelength of 635 nm. Is this correct? If not, please correct whichever is not proper.
    3. Please note: Excitation and emission wavelengths utilized during neurosurgery are not optimal wavelengths for fluorescence-guided surgery due to the presence of oxygenated and deoxygenated hemoglobin, which absorbs and scatters light. Thus, those wavelengths used in vivo do not necessarily need to be used for in vitro microscopy.
  4. My main point of contention is this: Why did you excite at 635 nm? By exciting at 635 nm, the nearly lowest fluorescence emission possible would be observed (mirror rule – the fluorescence intensity will be identical to the maximum of the absorption peak at which you are exciting – so, a small absorption peak generates a small fluorescence peak). Thus, your limit of detection is significantly artificially higher than had you excited at 405 nm and used a longpass/bandpass filter to collect all the emission from 635 nm and beyond. Exciting at 405 nm will result in the strongest fluorescence intensity at 720 nm because the absorbance at 405 nm is tremendously greater. As such, previous undetected fluorescence in low grade gliomas could/may now be detected because of achieving a significantly greater fluorescence intensity. Also, there would be no concerns of observing the other fluorescent dyes that have lower wavelength emission because you would be using a bandpass/longpass filter at 635 nm, and thereby primarily collecting Protoporphyrin IX emission. Of note, during microscopy studies, the laser power, gain, pinhole, contrast, and etc. (i.e., all settings) must remain identical between all specimens grades I-IV. To do the experiment properly, use the grade I for creating the baseline minimum fluorescence such that grade II-IV would only display higher fluorescence. Do not use the grade IV as the ceiling because it affords an artificially elevated baseline, for which any fluorescence beneath would not be detected. Thus, you could be inadvertently not visualizing the fluorescence because the baseline settings were initially set far above it.
  5. However, even after performing the crude experiment that I set forth in #4, the lower/no fluorescence from grades I-II may be the result of their lower metabolic activity, and not a function of whether 5-ALA crosses the BBB. This should be accounted for as a possibility if in fact #4 experiment still provides no fluorescence in grades I-II.
  6. Making claims about the BBB would require significant experiments to support such claims, and thus I would caution the authors to at most be speculative about 5-ALA not crossing the BBB. For example, “another possibility that could account for the low/no fluorescence in the grade I-II glioma could be the potential for 5-ALA to not cross the BBB”. However, as a small molecule under ~400 g/mol and with it adhering to Lipinski’s rule of 5, 5-ALA should have no problem of crossing the BBB. It may cross, but active efflux transporters may simply shuttle it back across the BBB.
  7. In summary, rewrite the introduction, perform experiment number #4, and revise your conclusions.